# Insulin-like Growth Factor-Binding Protein 2 in Severe Aortic Valve Stenosis and Pulmonary Hypertension: A Gender-Based Perspective

**DOI:** 10.3390/ijms25158220

**Published:** 2024-07-27

**Authors:** Elke Boxhammer, Vera Paar, Kristen Kopp, Sarah X. Gharibeh, Evelyn Bovenkamp-Aberger, Richard Rezar, Michael Lichtenauer, Uta C. Hoppe, Moritz Mirna

**Affiliations:** Department of Internal Medicine II, Division of Cardiology, Paracelsus Medical University of Salzburg, 5020 Salzburg, Austria

**Keywords:** aortic valve stenosis, gender, IGF-BP2, pulmonary hypertension, transcatheter aortic valve replacement

## Abstract

Severe aortic valve stenosis (AS) and pulmonary hypertension (PH) are life-threatening cardiovascular conditions, necessitating early detection and intervention. Recent studies have explored the role of Insulin-like Growth Factor-Binding Protein 2 (IGF-BP2) in cardiovascular pathophysiology. Understanding its involvement may offer novel insights into disease mechanisms and therapeutic targets for these conditions. A total of 102 patients (46 female, 56 male) with severe AS undergoing a transcatheter aortic valve replacement (TAVR) in a single-center study were classified using echocardiography tests to determine systolic pulmonary artery pressure (sPAP) and the presence (sPAP ≥ 40 mmHg) or absence (sPAP < 40 mmHg) of PH. Additionally, serial laboratory determinations of IGF-BP2 before, and at 24 h, 96 h, and 3 months after intervention were conducted in all study participants. Considering the entire cohort, patients with PH had significant and continuously higher serum IGF-BP2 concentrations over time than patients without PH. After subdivision by sex, it could be demonstrated that the above-mentioned results were only verifiable in males, but not in females. In the male patients, baseline IGF-BP2 levels before the TAVR was an isolated risk factor for premature death after intervention and at 1, 3, and 5 years post-intervention. The same was valid for the combination of male and echocardiographically established PH patients. The predictive role of IGF-BP2 in severe AS and concurrent PH remains unknown. A more profound comprehension of IGF-BP2 mechanisms, particularly in males, could facilitate the earlier consideration of the TAVR as a more effective and successful treatment strategy.

## 1. Introduction

In most cases, severe aortic valve stenosis (AS) is a disease of advanced age and is currently the most common valvular heart disease. AS is treated either surgically with a surgical aortic valve replacement (SAVR) or interventionally with a transcatheter aortic valve replacement (TAVR) [1]. SAVRs involve open-heart surgery and have long been the gold standard of treatment. However, TAVRs are less invasive and have emerged as a viable alternative, particularly for high-risk or inoperable patients, offering potential benefits in terms of reduced recovery time and lower procedural risks [2,3,4].

AS is primarily caused by a degenerative, calcific disease, characterized by the accumulation of calcium deposits on the aortic valve leaflets. This calcification leads to restricted valve mobility and impaired function, resulting in a narrowed valve orifice. The increased pressure gradient across the stenotic aortic valve places a significant burden on the left ventricle, which has to work harder to overcome the obstruction to pump blood into systemic circulation [5,6]. Over time, if untreated, this chronic pressure overload can lead to left ventricular hypertrophy and eventually heart failure. In the short or long term, however, not only the left heart but also the right heart is affected. The increased pressure in the left ventricle is transmitted backward into pulmonary circulation, causing elevated pulmonary arterial pressure. Prolonged exposure to high pressure in the pulmonary arteries induces the remodeling and constriction of the pulmonary vasculature, further increasing the resistance to the flow of blood and leading to pulmonary hypertension (PH). As a result, the right ventricle also has to generate a higher-pressure gradient, ultimately causing right ventricular hypertrophy and potentially leading to right-sided heart failure [1,7].

Studies have shown that the coexistence of PH in patients with severe AS is associated with increased mortality rates. The severity of PH has been correlated with worse outcomes, with higher systolic pulmonary artery pressures (sPAPs) observed in transthoracic echocardiography tests, which are indicative of a poorer prognosis. Additionally, the presence of right ventricular dysfunction, often attributed to the combined effect of severe AS and PH, has been linked to increased mortality and adverse cardiovascular events [8,9,10].

In recent years, there has been a growing interest in identifying cardiovascular biomarkers that can aid in the diagnosis, risk stratification, and management of various cardiac conditions including AS and PH. One such biomarker that has garnered attention is Insulin-like Growth Factor-Binding Protein 2 (IGF-BP2). IGF-BP2 is a protein that binds to Insulin-like Growth Factor 1 (IGF-1) and Insulin-like Growth Factor 2 (IGF-2), regulating their bioavailability and activity. Although initially studied in the context of cancer, emerging evidence suggests its potential predictive role in cardiovascular diseases [11,12].

The identification of reliable biomarkers such as IGF-BP2 could offer several advantages in the clinical management of severe AS stenosis-induced PH. Biomarkers can aid in early diagnosis and risk stratification, as well as serve as a tool for monitoring disease progression, and assessing treatment response [13,14,15]. Therefore, our study sought to evaluate the role of IGF-BP2 in patients with severe AS undergoing a TAVR in the presence or absence of PH. Serum levels of IGF-BP2 were determined in the patient’s blood immediately before the TAVR as well as 24 h, 96 h, and 3 months after the TAVR. The aim was not only to gain information about IGF-BP2 expression in the context of PH accompanying severe AS, but also to investigate any gender-specific differences regarding cardiovascular and all-cause mortality.

## 2. Results

### 2.1. Kaplan–Meier Curves

Primarily, Kaplan–Meier curves of the overall cohort (Figure 1A) and gender-specific cohort (male: Figure 1B, female: Figure 1C) were generated according to the presence or absence of pulmonary hypertension, which was determined using an sPAP cut-off value ≥ 40 mmHg. In the entire cohort, an increased sPAP value was associated with significantly increased mortality, excepting the 4-year survival timepoint.

Additionally, separate Kaplan–Meier curves were generated for male and female participants, based on the selected sPAP cut-off value. The log-rank test revealed a significant difference in premature death at each time point (1–5-year survival) for males. However, there were no statistically significant differences observed in long-term survival among female study participants.

The detailed statistical data analyses with log-rank tests and the number of patients at risk are shown in the corresponding Figure 1.

### 2.2. Study Cohort and Baseline Characteristics

The population included in this study comprised *n* = 102 patients who demonstrated severe AS on transthoracic echocardiography scans and were prioritized for a TAVR using a multidisciplinary decision-making process. The cohort included 56 men and 46 women, with an overall mean age of 82.3 ± 5.4 years. With the exception of typical gender-specific differences in height, body weight, and hemoglobin (HB), there were balanced patient characteristics between men and women. Significant differences with respect to gender distribution were seen for concomitant aortic valve regurgitation ≥ II° and/or additional mitral valve regurgitation ≥ II° pre-TAVR. Women had a significantly higher STS score in contrast to men. A detailed breakdown of the baseline characteristics is given in Table 1; echocardiographic details pre- and post-TAVR are demonstrated in Table 2.

### 2.3. IGF-BP2 Concentration in Overall Cohort and in Gender-Specific Cohort

IGF-BP2 levels were examined at various time points, and the results stratified by gender are presented in Figure 2 and Table 3.

Consistently, a significant increase in IGF-BP2 levels was observed in the first 24 h after the TAVR, both in the entire cohort (Figure 2A) and in the isolated subgrouping of men (Figure 2B) and women (Figure 2C). Subsequently, there was a subsequent decline in the investigated biomarker; this biomarker was particularly significant when compared between the post-interventional value at 24 h and at 3 months in the overall cohort and in the male gender.

At baseline, there were no significant differences between men and women in IGF-BP2 levels (*p* = 0.775). Following treatment, IGF-BP2 levels at 24 h and 96 h post-TAVR also did not differ in a statistically significant way (*p* = 0.249 and *p* = 0.977) between both genders. However, a notable difference emerged at the 3-month follow-up, with women demonstrating significantly higher IGF-BP2 levels compared to men (*p* = 0.010).

### 2.4. Correlation Analysis of sPAP and IGF-BP2 in Overall Cohort and in Gender-Specific Subdivision

A correlation analysis of the overall cohort and gender-specific cohort was conducted to investigate the relationship between sPAP and IGF-BP2 levels at various time points in patients with severe AS and PH, aiming to elucidate potential gender-specific differences and temporal dynamics in their association. The correlation analysis of the overall cohort (Table 4) revealed significant positive correlations between sPAP and IGF-BP2 levels across all time points: baseline (r = 0.287, *p* = 0.003), 24 h post-TAVR (r = 0.345, *p* = 0.001), 96 h post-TAVR (r = 0.379, *p* = 0.001), and 3 months post-TAVR (r = 0.275, *p* = 0.012). These findings suggest a consistent association between sPAP and IGF-BP2 levels throughout different stages of assessment, indicating a potential role of IGF-BP2 in the pathophysiology of pulmonary hypertension in severe aortic valve stenosis patients. The correlation analysis with respect to gender (Table 5: male; Table 6: female) reveals notable differences in the association between sPAP and IGF-BP2 levels with respect to gender. In males, there is a consistently stronger positive correlation between sPAP and IGF-BP2 levels across all time points: baseline (r = 0.345, *p* = 0.009), 24 h post-TAVR (r = 0.436, *p* = 0.002), 96 h post-TAVR (r = 0.429, *p* = 0.006), and 3 months post-TAVR (r = 0.313, *p* = 0.036). Conversely, in females, the correlations are notably weaker and not statistically significant across all time points. These findings suggest a consistent association between sPAP and IGF-BP2 levels in males, mirroring the patterns observed in the overall cohort, and implying a potentially crucial role of IGF-BP2 in the pathophysiology of PH in severe AS patients, particularly among males.

### 2.5. IGF-BP2 Concentration in Overall Cohort and in Gender-Specific Cohort Regarding Presence or Absence of PH

To assess the changes in serum IGF-BP2 concentration before and especially after TAVR implantation in relation to the presence or absence of PH, the data are graphically presented in the form of boxplots, as shown in Figure 3. Regardless of gender (Figure 3A), it could be demonstrated that the presence of PH was generally associated with a significantly increased IGF-BP2 concentration independent of the time of blood collection (baseline: *p* = 0.017; 24 h post-TAVR: *p* = 0.003; 96 h post-TAVR: *p* = 0.037; 3 months post-TAVR: *p* = 0.042). Additionally, it is interesting to note that within both the PH group and the non-PH group, there was a highly significant increase in IGF-BP2 concentration in the first 24 h post-TAVR (sPAP ≥ 40 mmHg: *p* = <0.001; sPAP < 40 mmHg: *p* = <0.001), respectively, and a significant decrease in serum concentration again in the further course from 24 h to 96 h (sPAP ≥ 40 mmHg: *p* = 0.032; sPAP < 40 mmHg: *p* = 0.021), respectively.

When compared to the overall cohort, similar boxplots of IGF-BP2 concentration trajectories were observed only in male study subjects (Figure 3B). Men with severe AS and evidence of concomitant, pre-interventional PH had consecutively and significantly higher IGF-BP2 levels, both before and after TAVR, than in male counterparts without previously established PH (baseline: *p* = 0.022; 24 h post-TAVR: *p* = 0.007; 96 h post-TAVR: *p* = 0.026; 3 months post-TAVR: *p* = 0.036). Both PH patients and non-PH patients showed a significant reduction in IGF-BP2 serum levels when comparing values 24 h vs. 3 months post-TAVR (sPAP ≥ 40 mmHg: *p* = 0.032; sPAP < 40 mmHg: *p* = 0.004).

In contrast to the boxplots of the total cohort or the male subjects, the boxplots of female patients did not show any significant differences in IGF-BP2 concentrations when comparing the PH vs. non-PH subsets (Figure 3C). Significant differences in terms of up-regulation were seen within the PH and non-PH groups, respectively, when comparing baseline IGF-BP2 concentration and 24-h post-TAVR values (sPAP ≥ 40 mmHg: *p* = 0.001; sPAP < 40 mmHg: *p* = 0.015).

### 2.6. Classification of Aortic Stenosis According to Généreux et al. [16] and IGF-BP2 Severity

To investigate the expression of IGF-BP2 with respect to the presence and extent of cardiac damage, patients were classified according to the stages defined by Généreux et al. [16], as mentioned above. The IGF-BP2 levels were analyzed separately (for the entire cohort), as well as for males and females over the known temporal course (Figure 4). As indicated by the baseline characteristics, only one female patient exhibited Stage 4 severity according to Généreux et al. [16], indicating extensive damage to the right cardiac system. Therefore, no median ± IQR could be detected here. From all three curve profiles, it becomes evident that a higher stage according to Généreux et al. [16] tends to correlate with higher plasma levels of IGF-BP2. Particularly, in the presence of Stage 4 severity, there was a marked increase in plasma levels observed in the overall cohort as well as in the male gender cohort.

### 2.7. Cox Hazard Regression Analysis: Males

The univariate and multivariable Cox hazard regression analyses with respect to male gender were analyzed (Table 7 and Appendix A). In males, IGF-BP2 pre-TAVR was an independent, significant risk factor for 1-, 3-, and 5-year mortality after a TAVR (1-year: *p* = 0.028; 3-years: *p* = 0.001; 5-years: *p* = 0.016).

### 2.8. Cox Hazard Regression Analysis: Male + sPAP ≥ 40 mmHg

Similarly, a Cox hazard regression analysis depending on the male gender and on the presence of PH (Table 8 and Appendix A), defined as sPAP ≥ 40 mmHg, demonstrated that elevated IGF-BP2 levels obtained immediately before the TAVR were also a significant risk factor for premature death 1-, 3-, and 5-years after the interventional valve replacement (1-year: *p* = 0.009; 3-years: 0.020; 5-years: *p* = 0.005).

### 2.9. Cox Hazard Regression Analysis: IGF-BP2 Baseline in Male Gender ± sPAP ≥ 40 mmHg

A further Cox regression analysis elucidates the association between IGF-BP2 levels at baseline and mortality outcomes across different time intervals, with a particular focus on males and sPAP levels ≥ 40 mmHg in males (Table 9). For 1-year mortality, higher baseline IGF-BP2 levels (>100 ng/mL to >600 ng/mL) exhibit increasingly higher hazard ratios, with statistically significant associations observed for levels >400 ng/mL, >500 ng/mL, and >600 ng/mL in the male cohort. When considering sPAP ≥ 40 mmHg in males, similar trends are noted, albeit with slightly attenuated hazard ratios. Similarly, for 3-year and 5-year mortality, elevated baseline IGF-BP2 levels (>100 ng/mL to >600 ng/mL) demonstrate progressively higher hazard ratios. Notably, when factoring in sPAP ≥ 40 mmHg in males, the associations remain consistent, albeit with some fluctuations in hazard ratios.

### 2.10. IGF-BP2 Concentration in Males with Respect to 1-, 3-, and 5-Year Survival Dependent upon Presence or Absence of PH

A comparison of serum IGF-BP2 levels in male patients with PH (sPAP ≥ 40 mmHg) in relation to survival or death at 1, 3, and 5 years is provided in Figure 5. The most important conclusion that could be drawn from this graphic is that there were no significant differences in IGF-BP2 serum concentrations between the two analyzed groups (survival vs. death) at any blood draw time point after 1 (Figure 5A), 3 (Figure 5B), or 5 (Figure 5C) years.

## 3. Material and Methods

### 3.1. Patient Population

The study population comprised 102 patients diagnosed with severe primary degenerative AS, who were scheduled to undergo TAVR between 1 January 2016 and 31 December 2018 at Paracelsus Medical University Hospital Salzburg. Echocardiography testing was available regarding PH before the procedure. Patients (*n* = 8) were excluded from the study if they had acute cardiac decompensation at the time of transthoracic echocardiography or TAVR, a bicuspid valve, complex congenital heart diseases, or a history suggestive of a pre-capillary component of PH. This included conditions such as chronic obstructive pulmonary disease (COPD) GOLD 4, idiopathic pulmonary arterial hypertension, chronic thromboembolic PH, interstitial lung disease, or rheumatologic diseases with pulmonary involvement, such as scleroderma or lupus erythematosus.

Data analyses were conducted adhering to the principles outlined in the Declaration of Helsinki and following good clinical practice guidelines. Approval for the study protocol was obtained from the local ethics committees of the Paracelsus Medical University Salzburg (415-E/1969/5-2016). Prior to participation, all patients provided written informed consent for their involvement in the study. Patients were followed up for a time period of up to five years with the primary endpoint of overall mortality.

### 3.2. Transthoracic Echocardiography

Transthoracic echocardiography testing was performed by four clinical investigators with more than 5 years of experience in cardiac ultrasound diagnostics using commonly used ultrasound devices (iE33 and Epiq 5; Philips Healthcare, Hamburg, Germany). The classification of severe AS was based on the current guidelines of the European Society for Cardiology (ESC) [17]. Severe AS was defined as an AV Vmax (maximal velocity over aortic valve) of ≥4.0 m/s, an AV dpmean (mean pressure gradient over aortic valve) ≥ 40 mmHg, and an aortic valve area ≤ 1.0 cm^2^. Patients with low-flow, low-gradient AS, and a stroke volume < 35 mL/m^2^ were excluded from the study, ensuring that the patient population analyzed consisted of individuals with high pressure gradients without a low-flow situation.

Left ventricular ejection fraction (LVEF) was calculated using Simpson’s method. Spectral and color Doppler images were utilized to assess the severity of mitral, aortic, and tricuspid valve regurgitation, categorized as minimal, mild (I), moderate (II), or severe (III). The maximum tricuspid regurgitant jet velocity (TRV) obtained through continuous wave Doppler imaging over the tricuspid valve was used to calculate pulmonary artery pressure (PAP) using the formula 4 × TRVmax^2^. To estimate the sPAP, which is critical for PH, the right atrial pressure (RAP) needed to be estimated. The RAP was determined based on the diameter of the inferior vena cava (IVC), corresponding to the central venous pressure. In the Salzburg and Linz cohorts, an IVC diameter ≥ 21 mm and a respiratory caliber fluctuation < 50% were assumed to have a RAP of 15 mmHg. An IVC diameter < 21 mm and a respiratory caliber fluctuation ≥ 50% resulted in a calculated RAP of 3 mmHg. Other scenarios not falling into the above categories were assigned an intermediate RAP value of 8 mmHg. The simplified Bernoulli equation (4 × TRVmax^2^) + RAP was used to determine the sPAP. An sPAP ≥ 40 mmHg was considered the cut-off value for diagnosing PH based on the current literature [17,18,19,20].

### 3.3. Classification of Aortic Stenosis Based on the Extent of Cardiac Damage

In addition to the commonly used echocardiographic classification of severe AS, Généreux et al. [16] described a novel staging classification in 2017 based on the presence and extent of cardiac damage. Stage 0 denotes the absence of any other cardiac damage. Stage 1 indicates left ventricular (LV) damage, characterized by LV hypertrophy (LV mass index > 95 g/m^2^ for women, >115 g/m^2^ for men), severe LV diastolic dysfunction (E/e_0_ > 14), or LV systolic dysfunction (LV ejection fraction < 50%). Stage 2 represents left atrial (LA) or mitral valve damage or dysfunction, identified by an enlarged LA (>34 mL/m^2^), the presence of atrial fibrillation, or moderate to severe mitral regurgitation. Stage 3 signifies pulmonary artery vasculature or tricuspid valve damage or dysfunction, evidenced by systolic pulmonary hypertension (systolic pulmonary arterial pressure ≥ 60 mm Hg) or moderate to severe tricuspid regurgitation. Finally, Stage 4 indicates right ventricular (RV) damage, marked by moderate to severe RV dysfunction.

### 3.4. Chronic Obstructive Pulmonary Disease (COPD) Definition

COPD was defined using the Global Initiative for Chronic Obstructive Lung Disease (GOLD) criteria valid at the time of the recruitment period [21], which include a post-bronchodilator forced expiratory volume in 1 s (FEV1)/forced vital capacity (FVC) ratio of <0.70. The severity of COPD was categorized based on FEV1 as follows:GOLD 1 (mild): FEV1 ≥ 80% predicted;GOLD 2 (moderate): 50% ≤ FEV1 < 80% predicted;GOLD 3 (severe): 30% ≤ FEV1 < 50% predicted;GOLD 4 (very severe): FEV1 < 30% predicted.

Patients with a known diagnosis of COPD, determined through their medical history and the use of inhalation therapies, were included in the COPD subgroup. The presence of COPD was considered when evaluating its impact on PH. However, patients with COPD GOLD 4 were excluded from the study, as previously mentioned.

### 3.5. TAVR

The decision to undergo a TAVR was determined through a collaborative approach involving a multidisciplinary team comprising cardiologists and cardiac surgeons. Specifically, all 102 patients underwent a TAVR using a transfemoral approach with the utilization of a CoreValve prosthesis (Medtronic, Dublin, Ireland).

Valve sizing was based on pre-procedural multimodal imaging assessments, including transthoracic echocardiography, computed tomography, and, where necessary, transesophageal echocardiography tests. The aortic annulus measurements, obtained from these imaging modalities, were crucial in selecting the appropriate valve size to ensure optimal valve positioning and function. The CoreValve Evolut R prostheses are available in various sizes (23 mm, 26 mm, 29 mm, and 34 mm), allowing for customization based on individual patient anatomy.

### 3.6. IGF-BP2 Analysis

Blood samples were collected under fasting conditions from the patients one day prior to the TAVR procedure and 24 h, 96 h, and 3 months after the TAVR, using a vacutainer blood collection system. The collected blood was processed using centrifugation to separate the plasma from the other components, and the plasma samples were subsequently frozen at −80 °C. This allowed the analysis of a total of 4 × 102 samples at similar time points and under the same conditions.

The plasma IGF-BP2 levels were measured using enzyme-linked immunosorbent assay (ELISA) kits. The ELISA kit used for this biomarker was IGF-BP2 (DY674) from R&D Systems (Minneapolis, MN, USA). The manufacturers’ instructions were followed to ensure proper reagent preparation.

In summary, the serum samples and standard proteins were loaded onto wells of ELISA plates (Nunc MaxiSorp flat-bottom 96-well plates, VWR International GmbH, Vienna, Austria) and incubated for two hours. The plates were then treated with a solution of Tween 20/PBS (Sigma Aldrich, St. Louis, MO, USA), followed by the addition of a biotin-labeled antibody. Another two-hour incubation followed. Subsequently, a washing process was performed, and a solution of streptavidin–horseradish–peroxidase was added to the wells. After adding tetramethylbenzidine (TMB; Sigma Aldrich, St. Louis, MO, USA), a color reaction was generated. The optical density of the samples was determined at 450 nm using an ELISA plate reader (iMark Microplate Absorbance Reader, Bio-Rad Laboratories, Vienna, Austria).

### 3.7. Statistical Analysis

Statistical analysis and graphical representations were conducted using SPSS (Version 25.0, SPSS Inc., Armonk, NY, USA). The normal distribution of variables was assessed using the Kolmogorov–Smirnov test. Normally distributed metric data were presented as mean ± standard deviation (SD) and analyzed using unpaired Student’s *t*-test. Non-normally distributed metric data were reported as median and interquartile range (IQR), and the Mann–Whitney U test was employed for statistical analysis. Categorical clinical data were presented as frequencies/percentages and compared using the chi-squared test. Friedman one-way ANOVA tests with pairwise comparison were utilized to evaluate changes in biomarker levels over time within the sPAP groups (<40 mmHg and ≥40 mmHg). The impact of echocardiographically-measured PH on patient survival (1- to 5-year survival) was visually depicted using Kaplan–Meier curves for the overall cohort and by gender. Correlation analyses were performed using Pearson’s correlation coefficient (parametric data) and Spearman’s rank correlation coefficient (non-parametric data) to determine the strength between sPAP and IGF-BP2 of different expression times to further variables. The univariate Cox regression analysis was conducted to identify potential influencing factors associated with 1, 3, and 5-year mortality. For better comparability, a z-transformation was applied to metric data. Subsequently, a multivariable Cox regression was performed to assess independent factors in predicting 1, 3, and 5-year mortality. Covariates identified in the univariate analysis (*p* ≤ 0.100) were included, and a backward variable elimination was performed. A *p*-value of <0.050 was considered statistically significant.

## 4. Discussion

### 4.1. Role of IGF-BP2 in PH

IGF-BP2 has emerged as a significant player in the pathogenesis of PH. For example, Yang et al. [22] demonstrated that IGF-BP2 was significantly increased in patients with pulmonary arterial hypertension (PAH), in contrast to lung-healthy patients. Similarly, the same working group was able to show that IGF-BP2 was an isolated risk factor for premature death in PAH patients. Also, in a pediatric patient population with PAH [23], the level of circulating IGF-BP2 was associated with worse clinical outcomes and survival. Similarly, evidence was provided by Guiot et al. [24] showing that patients with idiopathic pulmonary fibrosis (IPF) had significantly higher IGF-BP2 levels in contrast to healthy subjects. The pathomechanisms of IGF-BP2 in PH involve its dysregulation and subsequent impact on vascular remodeling, inflammation, and fibrosis in the pulmonary vasculature.

Vascular Remodeling: PH is characterized by structural changes in the pulmonary arteries, including the thickening of the vessel walls due to the abnormal proliferation of vascular smooth muscle cells (VSMCs). The overexpression of IGF-BP2 disrupts the balance between insulin-like growth factors (IGFs) and their receptors, leading to the increased bioavailability of IGFs. This, in turn, stimulates the proliferation of VSMCs, contributing to the thickening of pulmonary artery walls and the narrowing of the vessel lumen, which further elevates pulmonary artery pressure [22,25,26].

Inflammation: Inflammatory processes play a significant role in the development and progression of PH. IGF-BP2 has been found to modulate the expression of pro-inflammatory cytokines and chemokines, promoting inflammation in the pulmonary arteries. This inflammation contributes to endothelial dysfunction, vascular injury, and the recruitment of immune cells, perpetuating the pathogenic cycle [27,28].

Fibrosis: The excessive deposition of collagen and other extracellular matrix components in the pulmonary artery walls is a hallmark of PH-associated vascular remodeling. IGF-BP2 has been shown to stimulate fibroblast proliferation and collagen synthesis, promoting fibrotic changes in the pulmonary vasculature. This fibrosis further stiffens the vessel walls, exacerbating the increase in pulmonary artery pressure [24,29,30].

### 4.2. Role of IGF-BP2 in Severe AS

The role of IGF-BP2 in the context of severe AS is much less discussed in the current literature than PH. A study of 208 TAVR patients demonstrated that IGF-BP2 was significantly correlated with LVEF and with EUROSCORE but was also a relevant predictor of 30-day and 1-year survival after TAVR [15].

### 4.3. Role of IGF-BP2 in Severe AS and PH

In the current literature, the relationship between TAVRs and IGF-BP2 expression has already been investigated in scattered studies. However, to date, relevant data showing the association between IGF-BP2, severe AS, and concomitant PH are lacking.

The main conclusions that could be drawn from the results of our study are presented here (Figure 6):

TAVR patients with pre-interventional PH consistently had higher IGF-BP2 levels than the comparison group without PH. However, taking into account the respective patient gender, this observation could only be verified in the male gender.

IGF-BP2 was an isolated risk factor for premature death in males alone and males with echocardiographic evidence of PH after TAVR regardless of the respective time period of 1, 3, and 5 years.

IGF-BP2 was a relevant prognostic marker in the above-mentioned constellation at baseline, but changes in plasma concentrations during the course after TAVR were not associated with long-term prognosis.

The dysregulation of IGF-BP2 expression described above is likely to be a relevant pathomechanism in the development of PH as a concomitant disease of severe AS. This is supported by the continuously elevated IGF-BP2 levels in patients with non-invasive evidence of PH. Relevant comparative studies that have investigated this relationship are lacking in the current literature. Radiological criteria, such as the ratio of the pulmonary artery diameter to the body surface area (PA/BSA) have been used to detect PH; further, significantly different expression levels of IGF-BP2 have been observed to detect the presence or absence of PH as demonstrated in prior studies by our research group [31].

Gender-specific differences have been observed to be correlated with different PH causes, including prevalence, clinical presentation, response to treatment, and prognosis [32]. Several studies have suggested that females may have a higher prevalence of certain types of PH, such as PAH, compared to males. Additionally, there may be differences in disease progression and survival rates between genders [33]. In the present study, the male and female genders were equally affected by PH in severe AS and accordingly showed similar sPAP levels. IGF-BP2 expression levels, with the exception of the 3-month control after TAVR, also did not show any significant differences when comparing sexes. However, in our study, it was clearly observed that men without PH had significantly lower baseline IGF-BP2 levels and that IGF-BP2 was a relevant prognostic marker for survival after TAVR in men alone. The reasons for this are manifold. For example, hormonal regulation plays a significant role in regulating the expression and activity of IGF-BP2. Hormones such as estrogen and testosterone, which are present in different concentrations in males and females, can influence IGF-BP2 levels. Estrogen, for instance, has been shown to up-regulate IGF-BP2 expression in certain tissues, while testosterone may have the opposite effect [34]. Gender-specific differences in IGF-BP2 expression have also been associated with variations in metabolism and body composition. In some studies, higher IGF-BP2 levels in males have been linked to greater lean body mass and lower body fat compared to females [35,36]. This relationship can be demonstrated in this study by an association between IGF-BP2 plasma levels and BMI or body weight isolated within the male gender (compare correlation analysis in Table 4 and Table 5). The interaction between IGF-BP2 and the endothelium described in the literature should be mentioned. In this context, it has been repeatedly reported that IGF-BP2 is not only a relevant factor for angiogenesis, but also a relevant stimulator for the proliferation of endothelial cells [25,37,38]. The extent to which this mechanism may lead to continuous pathological vasoconstriction, hypertrophy of the vessel walls, and ultimately to chronic inflammation of the endothelium, especially in males, needs to be investigated. Finally, the dysfunction of the right ventricle before TAVR [39] (evident in the classification of aortic stenosis according to Généreux presented in Figure 4) also has an influence on IGF-BP2; however, this finding has to be substantiated by further prospective studies.

## 5. Limitation

The small number of participants in a single center (*n* = 102), further divided into subgroups based on gender and the presence of PH, represents a primary limitation of this study. Additionally, technical challenges inherent in echocardiographic and laboratory assessments can lead to misclassifications, despite examinations being conducted by skilled clinical investigators and laboratory technicians.

We recognize the potential limitations of our method for defining PH, which relied on echocardiographic estimates of sPAP. While the use of TRVmax is recommended by guidelines, the clinical practice often uses sPAP for determining non-invasive PH. This approach aligns with the comparative literature, as many non-invasive studies on this topic also rely on sPAP. Therefore, for consistency and better comparability with existing research, we used sPAP throughout our study. However, we acknowledge that this method involves assumptions about RAP, based on the patient’s estimated filling status. These discrete values can introduce variability and potential bias, possibly affecting patient classification in our study.

While we did not include detailed functional pulmonary testing, we do have data regarding the presence or absence of COPD in our patients. This information provides some insight into the pulmonary status of our cohort, but we acknowledge that comprehensive pulmonary function testing would have offered a more complete assessment. This limitation should be considered when interpreting our results.

Furthermore, we did not perform invasive right heart catheterization, the gold standard for accurately diagnosing the type of PH (pre-capillary versus post-capillary). This procedure is no longer routinely conducted before transcatheter aortic valve replacements (TAVRs), meaning our cohort may not exclusively include patients with left heart-related, post-capillary PH.

## 6. Conclusions

Research into the role of IGF-BP2 in severe AS and concomitant PH is far from complete. A deeper understanding of the mechanisms related to IGF-BP2 in particular could pave the way to establish TAVRs earlier, if necessary, especially in the male gender, to treat the combination of these cardiovascular diseases even more effectively and successfully.

## Figures and Tables

**Figure 1 ijms-25-08220-f001:**
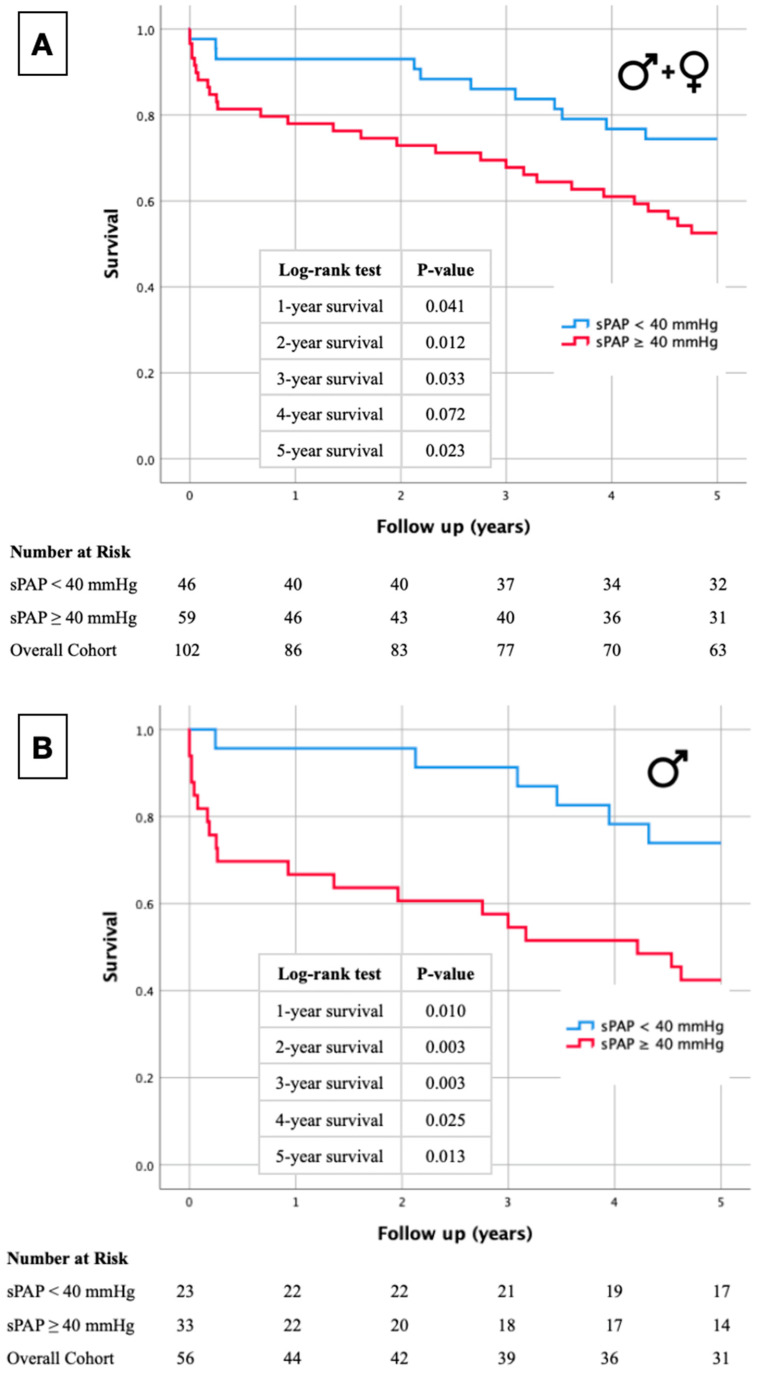
Kaplan–Meier curves with corresponding numbers at risk and annual log-rank tests for detection of 1- to 5-year survival in overall cohort (**A**) and with dependence on gender ((**B**) male; (**C**) female).

**Figure 2 ijms-25-08220-f002:**
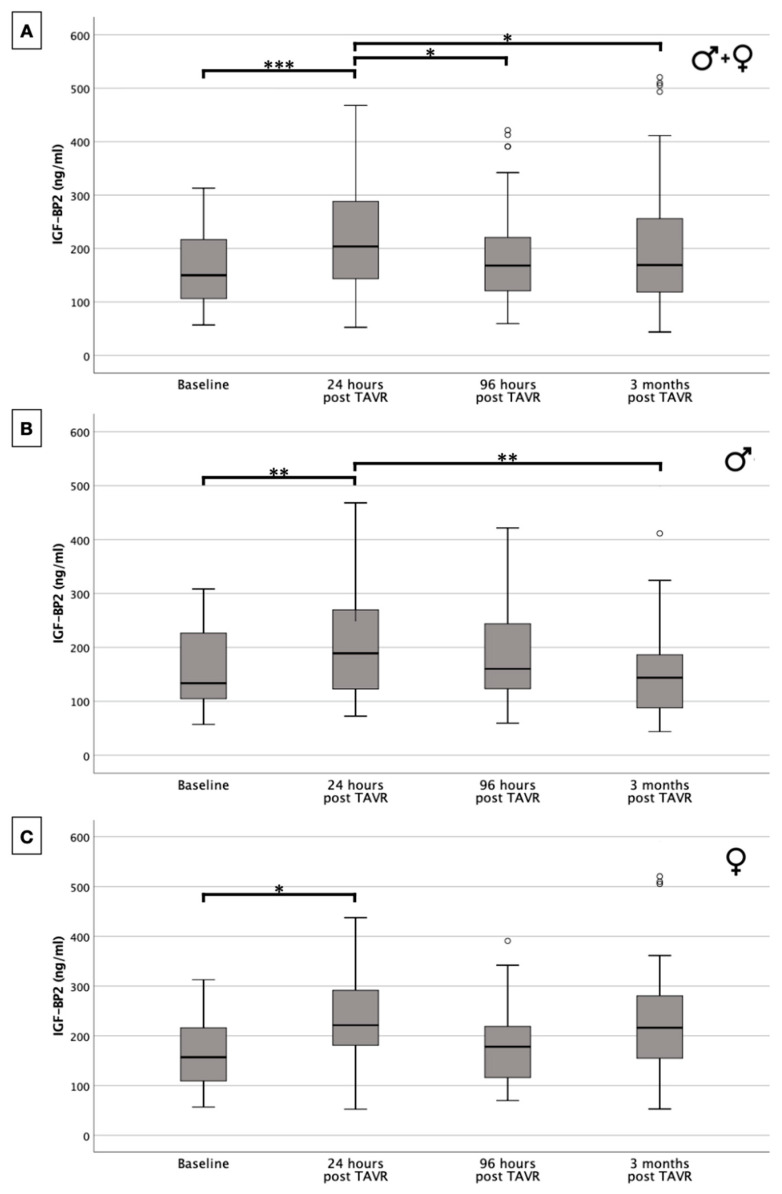
Serum concentration of IGF-BP2 in overall patient cohort (**A**) and with dependence on gender ((**B**) male; (**C**) female). * *p* ≤ 0.05; ** *p* ≤ 0.01; *** *p* ≤ 0.001. o = outlier.

**Figure 3 ijms-25-08220-f003:**
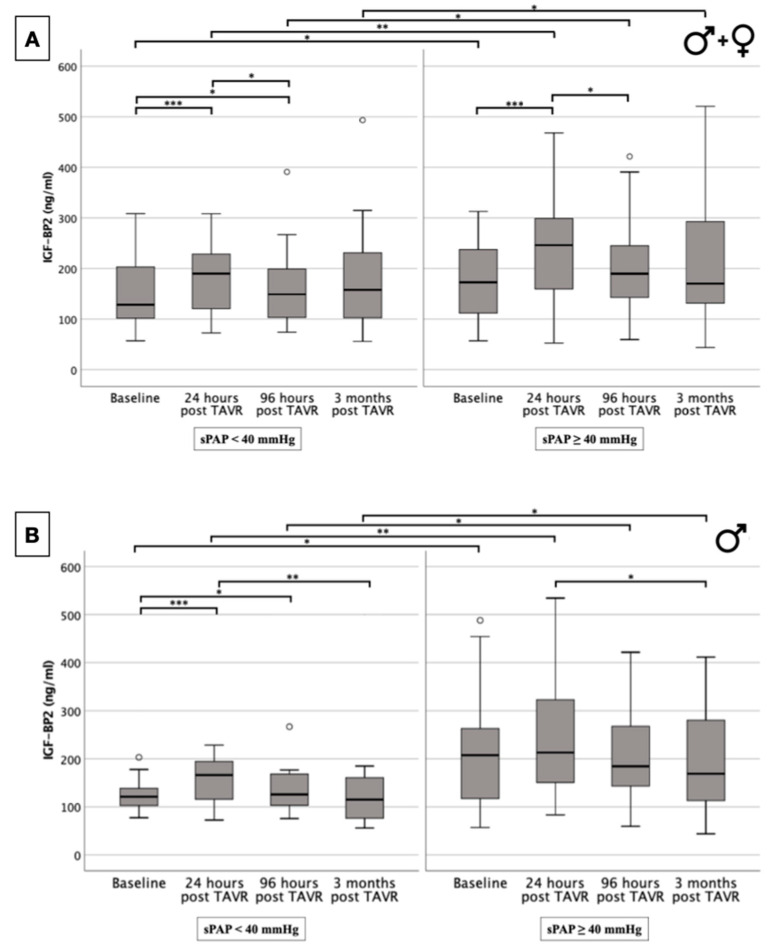
Serum concentration of IGF-BP2 in overall patient cohort (**A**) and with dependence on gender ((**B**) male; (**C**) female) with an sPAP < 40 mmHg and ≥ 40 mmHg. * *p* ≤ 0.05; ** *p* ≤ 0.01; *** *p* ≤ 0.001. o = outlier.

**Figure 4 ijms-25-08220-f004:**
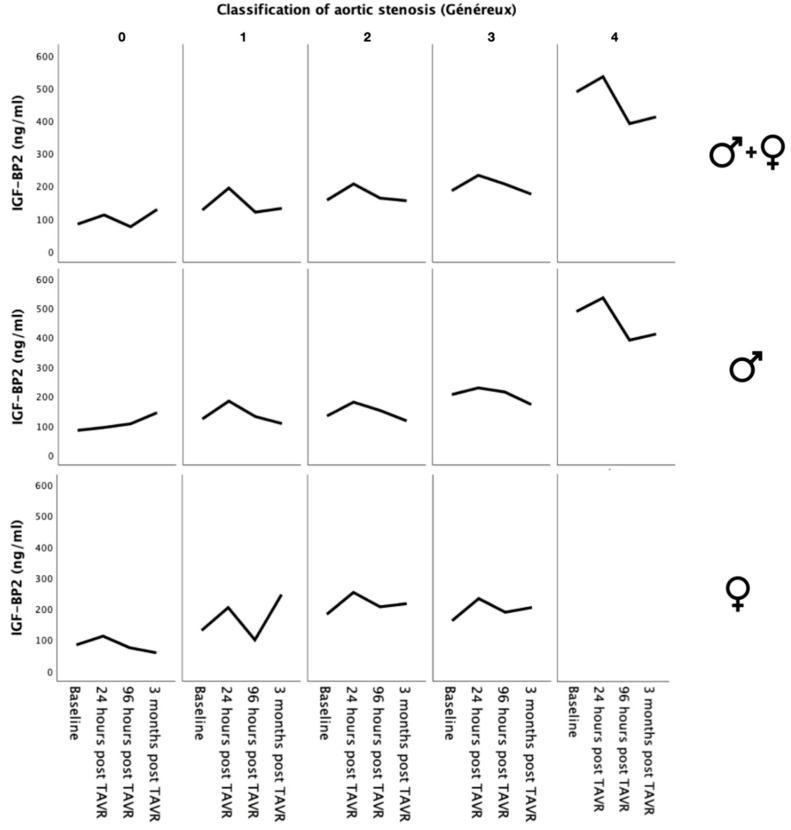
IGF-BP2 levels of overall cohort and male and female patients over time regarding different stages of aortic stenosis classified according to Généreux et al. [16].

**Figure 5 ijms-25-08220-f005:**
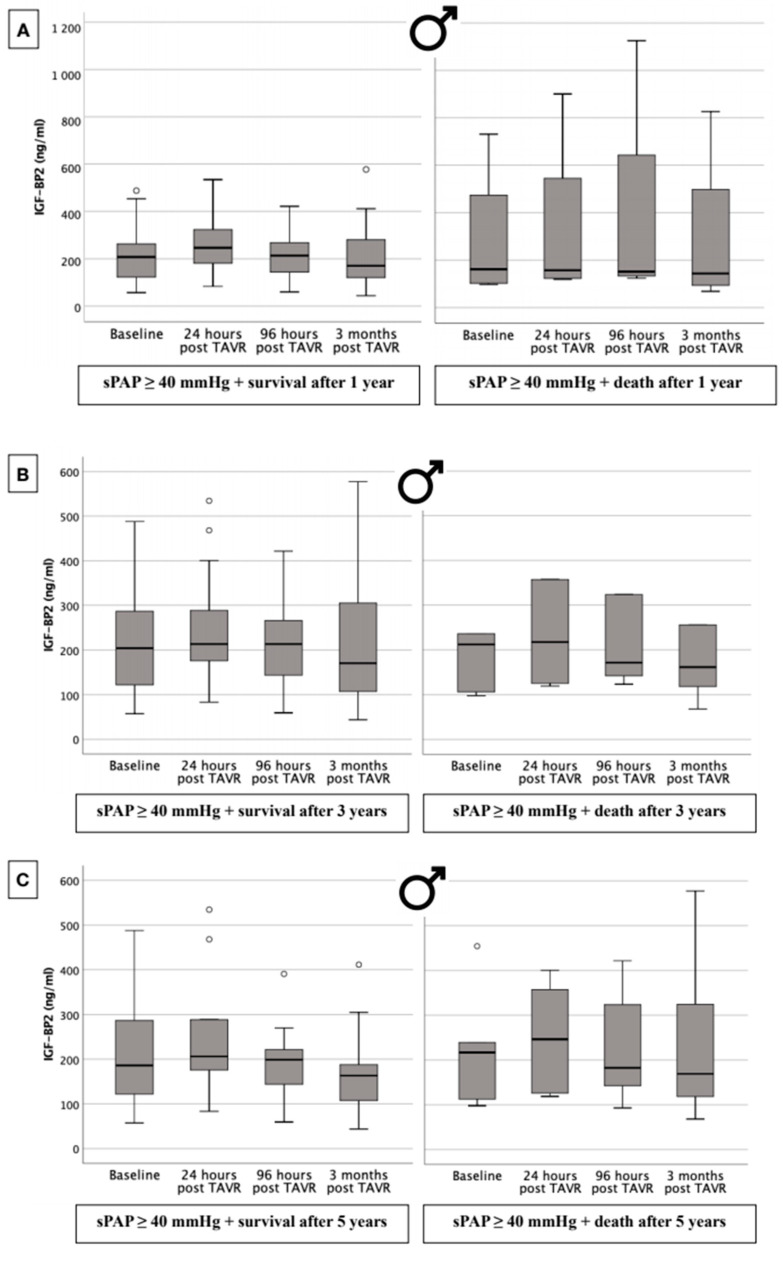
Serum concentration of IGF-BP2 in male patient cohort with an sPAP ≥ 40 mmHg regarding survival or death after 1 (**A**), 3 (**B**), and 5 years (**C**) after TAVR. o = outlier.

**Figure 6 ijms-25-08220-f006:**
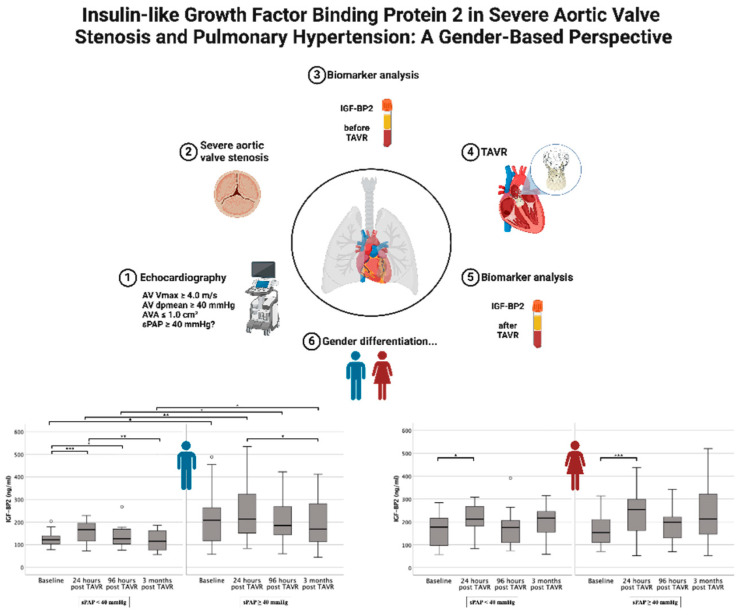
Graphic summary of the main findings of the study (Created with BioRender.com). Image material of CoreValve™ Evolut™ was kindly provided by **©** Medtronic Inc. * *p* ≤ 0.05; ** *p* ≤ 0.01; *** *p* ≤ 0.001; o = outlier.

**Table 1 ijms-25-08220-t001:** Baseline characteristics of study cohort.

	Total	Men	Women	*p*-Value
**No. (%)**				
Total	102 (100.0)	56 (54.9)	46 (45.1)	0.215
Age				
60–69	4 (3.9)	3 (5.4)	1 (2.2)	0.410
70–79	22 (21.6)	15 (26.8)	7 (15.2)	0.158
80–89	68 (66.7)	35 (62.5)	33 (71.7)	0.325
≥90	8 (7.8)	3 (5.4)	5 (10.9)	0.303
BMI				
<18.5	1 (1.0)	0 (0.0)	1 (2.2)	0.879
18.5–24.9	46 (45.1)	25 (44.6)	21 (45.7)	0.794
25.0–29.9	37 (36.3)	23 (41.1)	14 (30.4)	0.719
30.0–34.9	13 (12.7)	5 (8.9)	8 (17.4)	0.038
35.0–39.9	5 (4.9)	3 (5.4)	2 (4.3)	0.072
≥40.0	0 (0.0)	0 (0.0)	0 (0.0)	1.000
NYHA ≥ III	77 (75.5)	41 (73.2)	36 (78.3)	0.520
Diabetes mellitus	28 (27.5)	13 (23.2)	15 (32.6)	0.290
Arterial hypertension	85 (83.3)	47 (83.9)	38 (82.6)	0.859
CVD	62 (60.8)	36 (64.3)	26 (56.5)	0.424
Previous myocardial infarction	4 (3.9)	2 (3.6)	2 (4.3)	0.841
Atrial fibrillation	29 (28.4)	13 (23.2)	16 (34.8)	0.197
Previous cardiac surgery	3 (2.9)	3 (5.4)	0 (0.0)	0.111
Pacemaker (before TAVR)	5 (4.9)	3 (5.4)	2 (4.3)	0.814
Malignancy	24 (23.5)	15 (26.8)	9 (19.6)	0.392
Stroke (before TAVR)	9 (8.8)	5 (8.9)	4 (8.7)	0.967
PAOD	10 (9.8)	4 (7.1)	6 (13.0)	0.319
COPD	8 (7.8)	5 (8.9)	3 (8.7)	0.653
COPD Gold 1	2 (2.0)	1 (1.8)	1 (2.2)	0.888
COPD Gold 2	3 (2.9)	1 (1.8)	2 (4.3)	0.446
COPD Gold 3	3 (2.9)	3 (5.4)	0 (0.0)	0.111
COPD Gold 4	0 (0.0)	0 (0.0)	0 (0.0)	1.000
Pacemaker (after TAVR)	14 (13.7)	6 (10.7)	8 (17.4)	0.771
Vascular complications	11 (10.8)	6 (10.7)	5 (10.9)	0.980
Stroke (after TAVR)	0 (0.0)	0 (0.0)	0 (0.0)	1.000
Classification of aortic stenosis				
(Généreux et al. [16])				
Stage 0	7 (6.9)	5 (8.9)	2 (4.3)	0.363
Stage 1	12 (11.8)	5 (8.9)	7 (15.2)	0.327
Stage 2	44 (43.1)	24 (42.9)	20 (43.5)	0.950
Stage 3	33 (32.4)	17 (30.4)	16 (34.8)	0.635
Stage 4	6 (5.9)	5 (8.9)	1 (2.2)	0.149
**Mean ± SD**				
Age (years)	82.3 ± 5.4	81.5 ± 5.7	83.3 ± 4.9	0.100
Height (cm)	167.9 ± 8.7	173.6 ± 5.4	160.4 ± 6.0	<0.001
Weight (kg)	73.9 ± 13.1	78.9 ± 13.7	67.4 ± 9.0	<0.001
BMI (kg/m^2^)	26.2 ± 4.0	26.1 ± 4.4	26.3 ± 3.5	0.812
**Median ± IQR**				
STS score	2.6 ± 1.9	2.0 ± 1.2	3.5 ± 2.3	<0.001
Creatinine (mg/dL)	1.0 ± 0.5	1.1 ± 0.5	1.0 ± 0.4	0.327
BNP (pg/mL)	1591.0 ± 3041.3	1459.5 ± 3482.2	1591.0 ± 2324.1	0.543
HK (%)	38.9 ± 7.0	40.3 ± 8.0	37.8 ± 5.4	0.137
HB (g/dL)	13.0 ± 2.4	13.5 ± 2.9	12.7 ± 1.8	0.041
CK (U/L)	78.0 ± 79.0	79.5 ± 94.5	74.0 ± 57.0	0.289

BMI: body mass index; CVD: cardiovascular disease; PAOD: peripheral arterial occlusive disease; COPD: chronic obstructive pulmonary disease; TAVR: transcatheter aortic valve replacement; eGFR: estimated glomerular filtration rate; BNP: brain natriuretic peptide; HK: hematocrit; HB: hemoglobin; CK: creatine kinase.

**Table 2 ijms-25-08220-t002:** Echocardiographic criteria of study cohort pre- and post-TAVR.

	Total	Men	Women	*p*-Value
**Echocardiography pre-TAVR**				
**No. (%)**				
LVEF				
≤30	6 (5.9)	4 (7.2)	2 (4.3)	0.164
31–54	30 (29.4)	14 (25.0)	16 (34.8)	0.150
≥55	66 (64.7)	38 (67.8)	28 (60.9)	0.499
sPAP				
≥40	59 (57.8)	33 (58.9)	26 (56.5)	0.807
≥45	47 (46.1)	25 (44.6)	22 (47.8)	0.748
≥50	32 (31.4)	16 (28.6)	16 (34.8)	0.501
≥60	20 (19.6)	13 (23.2)	7 (15.2)	0.311
AVI ≥ II°	27 (26.4)	16 (28.6)	11 (23.9)	0.036
MVI ≥ II°	39 (38.2)	17 (30.4)	22 (47.8)	0.026
TVI ≥ II°	27 (26.5)	12 (21.4)	15 (34.9)	0.109
**Mean ± SD**				
LVEF (%)	53.4 ± 10.1	53.1 ± 11.6	53.7 ± 7.7	0.768
LVEDD (mm)	47.0 ± 6.1	47.9 ± 5.8	45.7 ± 6.2	0.101
AV Vmax (m/s)	4.4 ± 0.6	4.4 ± 0.4	4.4 ± 0.7	0.803
AV dpmax (mmHg)	81.2 ± 20.1	82.4 ± 14.7	79.8 ± 24.9	0.565
AV dpmean (mmHg)	49.3 ± 11.5	49.2 ± 8.5	49.50 ± 14.3	0.896
TAPSE (mm)	22.1 ± 3.7	22.5 ± 4.2	21.7 ± 3.2	0.348
sPAP (mmHg)	43.2 ± 19.6	44.0 ± 20.3	42.2 ± 18.9	0.654
**Echocardiography post-TAVR (before admission)**				
**No. (%)**				
PVL ≥ II°	17 (16.7)	10 (17.8)	7 (15.2)	0.374
MVI ≥ II°	37 (36.3)	16 (28.6)	21 (45.7)	0.076
TVI ≥ II°	28 (27.5)	13 (23.2)	15 (32.6)	0.290
**Mean ± SD**				
LVEF (%)	54.1 ± 7.5	53.7 ± 8.3	54.5 ± 6.5	0.642
LVEDD (mm)	46.3 ± 6.1	46.2 ± 6.4	46.4 ± 6.5	0.855
AV Vmax (m/s)	1.9 ± 0.4	1.9 ± 0.5	1.9 ± 0.4	0.682
AV dpmax (mmHg)	16.4 ± 7.5	17.0 ± 8.6	15.4 ± 5.4	0.363
AV dpmean (mmHg)	8.3 ± 3.6	8.6 ± 4.0	7.9 ± 3.0	0.398
TAPSE (mm)	21.4 ± 4.3	21.8 ± 4.4	20.9 ± 4.2	0.343
sPAP (mmHg)	40.8 ± 16.5	42.6 ± 14.7	39.2 ± 18.1	0.388

LVEF: left ventricular ejection fraction; sPAP: systolic pulmonary artery pressure; TAVR: transcatheter aortic valve replacement; LVEDD: left ventricular end-diastolic diameter at diastole; AV Vmax: maximal velocity over aortic valve; AV dpmax: maximal pressure gradient over; AV dpmean: mean pressure gradient over aortic valve; TAPSE: tricuspid annular plane systolic excursion; AVI: aortic valve insufficiency; MVI: mitral valve insufficiency; TVI: tricuspid valve insufficiency; PVL: paravalvular leak.

**Table 3 ijms-25-08220-t003:** Tabular overview of the IGF-BP2 values recorded over time and the number of measurements.

	Total	Men	Women	*p*-Value
**Median ± IQR** **No.**				
IGF-BP2 baseline (ng/mL)	149.8 ± 112.6 102/102	133.4 ± 127.2 56/56	157.0 ± 111.5 46/46	0.775
IGF-BP2 post 24 h (ng/mL)	203.8 ± 145.7 99/102	189.0 + 148.9 54/56	221.4 ± 132.9 45/46	0.249
IGF-BP2 post 96 h (ng/mL)	168.1 ± 102.5 99/102	160.2 ± 131.5 54/56	178.1 ± 107.0 45/46	0.977
IGF-BP2 post 3 months (ng/mL)	169.0 ± 148.2 91/102	143.7 ± 101.1 47/56	216.4 ± 148.4 44/46	0.010

IGF-BP2: Insulin-like Growth Factor-Binding Protein 2.

**Table 4 ijms-25-08220-t004:** Correlation analysis (sPAP and IGF-BP2) in overall cohort. * *p* ≤ 0.05; ** *p* ≤ 0.01; *** *p* ≤ 0.001.

	sPAP		IGF-BP2 Baseline		IGF-BP2 Post 24 h		IGF-BP2 Post 96 h		IGF-BP2 Post 3 Months	
	r	*p*	r	*p*	r	*p*	r	*p*	r	*p*
Age	0.039	0.773	0.166	0.222	0.272	0.056	0.256	0.116	0.413 **	0.005
Height	−0.100	0.478	−0.036	0.798	−0.062	0.679	−0.058	0.737	−0.048	0.760
Weight	−0.197	0.152	−0.376 **	0.005	−0.407 **	0.004	−0.365 *	0.026	−0.358 *	0.017
BMI	−0.165	0.237	−0.369 **	0.007	−0.397 **	0.006	−0.333 *	0.047	−0.356 *	0.019
NYHA	0.357	0.112	0.178	0.441	0.231	0.326	0.351	0.153	0.430	0.110
STS score	0.209	0.123	0.209	0.122	0.361 *	0.010	0.173	0.291	0.395 **	0.007
Diabetes mellitus	−0.240	0.075	−0.110	0.420	−0.114	0.432	−0.088	0.593	−0.050	0.743
Arterial hypertension	0.121	0.376	0.143	0.293	0.174	0.228	0.139	0.399	−0.060	0.693
CVD	−0.206	0.128	0.032	0.813	0.027	0.852	−0.180	0.272	−0.185	0.224
Previous myocardial infarction	0.084	0.541	0.292 *	0.029	0.297 *	0.036	0.289	0.074	0.244	0.107
Atrial fibrillation	0.243	0.072	0.207	0.126	0.230	0.107	0.225	0.168	0.191	0.209
Previous cardiac surgery	0.123	0.367	−0.120	0.378	−0.054	0.707	0.072	0.663	0.075	0.626
Pacemaker (before TAVR)	−0.002	0.986	0.292 *	0.029	0.255	0.074	0.186	0.257	0.268	0.078
Malignancy	−0.326 *	0.014	0.111	0.415	−0.065	0.654	−0.083	0.613	0.235	0.120
Stroke (before TAVR)	−0.101	0.459	−0.019	0.887	−0.012	0.937	−0.058	0.726	−0.151	0.322
PAOD	−0.024	0.836	0.047	0.730	−0.166	0.248	−0.077	0.641	0.024	0.875
COPD	0.146	0.284	−0.141	0.299	−0.095	0.513	−0.207	0.207	−0.185	0.223
LVEF	−0.193	0.153	−0.092	0.500	−0.138	0.339	−0.036	0.829	−0.011	0.941
LVEDD	0.012	0.934	−0.026	0.857	0.114	0.455	0.049	0.785	0.242	0.128
AV Vmax	−0.150	0.330	−0.055	0.725	−0.084	0.606	−0.098	0.600	−0.100	0.563
AV dpmax	−0.130	0.388	0.053	0.725	−0.119	0.446	−0.196	0.282	−0.190	0.253
AV dpmean	−0.102	0.490	−0.058	0.695	−0.198	0.198	−0.269	0.136	−0.225	0.162
TAPSE	−0.189	0.214	−0.114	0.455	−0.064	0.697	0.028	0.885	−0.309	0.075
sPAP	1.000	-	0.345 **	0.009	0.436 **	0.002	0.429 **	0.006	0.313 *	0.036
AVI ≥ II°	−0.165	0.329	0.089	0.599	0.200	0.271	0.218	0.255	0.080	0.659
MVI ≥ II°	0.147	0.286	0.185	0.175	0.211	0.146	0.202	0.218	0.080	0.604
TVI ≥ II°	0.417 **	0.002	0.183	0.186	0.213	0.146	0.265	0.108	0.082	0.602
Creatinine	0.171	0.208	0.536 ***	<0.001	0.442 **	0.001	0.289	0.074	0.465 **	0.001
BNP	0.458 **	0.002	0.451 **	0.002	0.343 *	0.026	0.248	0.158	0.154	0.370
HK	−0.413 **	0.002	−0.307 *	0.021	−0.430 **	0.002	−0.428 **	0.007	−0.356 *	0.016
HB	−0.433 **	0.001	−0.445 **	0.001	−0.507 ***	<0.001	−0.468 **	0.003	−0.384 **	0.009
CK	−0.073	0.616	−0.006	0.966	0.025	0.871	−0.066	0.710	0.197	0.230
IGF-BP2 baseline	0.345 **	0.009	1.000	-	0.870 ***	<0.001	0.738 ***	<0.001	0.842 ***	<0.001
IGF-BP2 post 24 h	0.436 **	0.002	0.870 ***	<0.001	1.000	-	0.876 ***	<0.001	0.833 ***	<0.001
IGF-BP2 post 96 h	0.429 **	0.006	0.738 ***	<0.001	0.876 ***	<0.001	1.000	-	0.837 ***	<0.001
IGF-BP2 post 3 months	0.313 *	0.036	0.842 ***	<0.001	0.833 ***	<0.001	0.837 ***	<0.001	1.000	-

sPAP: systolic pulmonary artery pressure; IGF-BP2: Insulin-like Growth Factor-Binding Protein 2; BMI: body mass index; CVD: cardiovascular disease; PAOD: peripheral arterial occlusive disease; COPD: chronic obstructive pulmonary disease; LVEF: left ventricular ejection fraction; sPAP: systolic pulmonary artery pressure; TAVR: transcatheter aortic valve replacement; LVEDD: left ventricular end-diastolic diameter at diastole; AV Vmax: maximal velocity over aortic valve; AV dpmax: maximal pressure gradient over; AV dpmean: mean pressure gradient over aortic valve; TAPSE: tricuspid annular plane systolic excursion; AVI: aortic valve insufficiency; MVI: mitral valve insufficiency; TVI: tricuspid valve insufficiency; eGFR: estimated glomerular filtration rate; BNP: brain natriuretic peptide; HK: hematocrit; HB: hemoglobin; CK: creatine kinase.

**Table 5 ijms-25-08220-t005:** Correlation analysis (sPAP and IGF-BP2) in male gender. * *p* ≤ 0.05; ** *p* ≤ 0.01; *** *p* ≤ 0.001.

	sPAP		IGF-BP2 Baseline		IGF-BP2 Post 24 h		IGF-BP2 Post 96 h		IGF-BP2 Post 3 Months	
	r	*p*	r	*p*	r	*p*	r	*p*	r	*p*
Age	0.093	0.354	0.140	0.161	0.133	0.205	0.236 *	0.046	0.394 ***	<0.001
Gender	−0.001	0.989	−0.028	0.777	0.121	0.251	−0.003	0.978	0.284 **	0.009
Height	0.133	0.201	0.112	0.283	−0.006	0.956	0.043	0.743	−0.120	0.301
Weight	−0.054	0.598	−0.228 *	0.025	−0.291 **	0.006	−0.278 *	0.023	−0.325 **	0.003
BMI	−0.161	0.121	−0.311 **	0.002	−0.327 **	0.002	−0.309 *	0.012	−0.277 *	0.015
NYHA	0.186	0.196	0.091	0.531	0.199	0.176	0.143	0.360	0.369 *	0.023
STS score	0.192	0.053	0.107	0.282	0.133	0.205	0.171	0.152	0.436 ***	<0.001
Diabetes mellitus	0.053	0.599	−0.035	0.729	−0.002	0.983	0.049	0.680	−0.098	0.377
Arterial hypertension	0.054	0.592	−0.025	0.806	0.060	0.567	−0.006	0.958	−0.073	0.514
CVD	−0.125	0.211	0.031	0.754	0.072	0.493	−0.111	0.353	−0.211	0.055
Previous myocardial infarction	−0.003	0.979	0.285 **	0.004	0.157	0.136	0.274 *	0.020	0.145	0.189
Atrial fibrillation	0.316 **	0.001	0.064	0.521	0.064	0.544	0.055	0.649	0.151	0.173
Previous cardiac surgery	0.102	0.309	−0.096	0.339	−0.061	0.562	0.060	0.617	0.016	0.883
Pacemaker (before TAVR)	−0.033	0.740	0.160	0.109	0.153	0.147	0.006	0.961	0.195	0.077
Malignancy	−0.160	0.107	0.032	0.748	−0.062	0.558	−0.066	0.581	0.118	0.288
Stroke (before TAVR)	−0.102	0.306	−0.039	0.699	−0.016	0.881	−0.027	0.820	−0.004	0.974
PAOD	−0.028	0.780	0.041	0.679	−0.061	0.562	0.019	0.873	0.108	0.333
COPD	−0.015	0.882	−0.022	0.824	−0.084	0.425	−0.108	0.368	−0.145	0.192
LVEF	−0.125	0.214	−0.117	0.247	−0.163	0.124	−0.019	0.876	−0.002	0.986
LVEDD	−0.070	0.510	−0.007	0.949	0.046	0.683	−0.034	0.797	0.047	0.693
AV Vmax	−0.084	0.453	−0.052	0.644	−0.080	0.498	0.009	0.945	0.033	0.791
AV dpmax	−0.089	0.410	0.025	0.820	−0.114	0.312	−0.033	0.800	0.043	0.720
AV dpmean	−0.044	0.683	−0.036	0.738	−0.045	0.688	−0.047	0.716	0.021	0.855
TAPSE	−0.270 *	0.015	0.041	0.720	−0.062	0.607	0.053	0.705	−0.144	0.256
sPAP	1.000	-	0.287 **	0.003	0.345 **	0.001	0.379 **	0.001	0.275 *	0.012
AVI ≥ II°	−0.006	0.965	0.071	0.575	0.253	0.055	0.161	0.279	0.200	0.144
MVI ≥ II°	0.173	0.089	0.147	0.149	0.132	0.220	0.216	0.075	0.213	0.060
TVI ≥ II°	0.418 ***	<0.001	0.127	0.216	0.150	0.166	0.239 *	0.049	0.119	0.301
Creatinine	0.233 *	0.018	0.464 ***	<0.001	0.351 **	0.001	0.291 *	0.013	0.395 ***	<0.001
BNP	0.366 **	0.001	0.431 ***	<0.001	0.333 **	0.004	0.272 *	0.034	0.185	0.131
HK	−0.344 ***	< 0.001	−0.242 *	0.014	−0.378 ***	<0.001	−0.352 **	0.002	−0.328 **	0.002
HB	−0.332 **	0.001	−0.330 **	0.001	−0.384 ***	<0.001	−0.377 **	0.001	−0.360 **	0.001
CK	−0.046	0.662	0.009	0.933	0.017	0.882	−0.062	0.622	0.085	0.468
IGF-BP2 baseline	0.287 **	0.003	1.000	-	0.776 ***	<0.001	0.735 ***	<0.001	0.691 ***	<0.001
IGF-BP2 24 h post-TAVR	0.345 ***	0.001	0.776 ***	<0.001	1.000	-	0.799 ***	<0.001	0.657 ***	<0.001
IGF-BP2 96 h post-TAVR	0.379 ***	0.001	0.735 ***	<0.001	0.799 ***	<0.001	1.000	-	0.734 ***	<0.001
IGF-BP2 3 months post-TAVR	0.275 *	0.012	0.691 ***	<0.001	0.657 ***	<0.001	0.734 ***	<0.001	1.000	-

sPAP: systolic pulmonary artery pressure; IGF-BP2: Insulin-like Growth Factor-Binding Protein 2; BMI: body mass index; CVD: cardiovascular disease; PAOD: peripheral arterial occlusive disease; COPD: chronic obstructive pulmonary disease; LVEF: left ventricular ejection fraction; sPAP: systolic pulmonary artery pressure; TAVR: transcatheter aortic valve replacement; LVEDD: left ventricular end-diastolic diameter at diastole; IVSd: interventricular septal thickness at diastole; AV Vmax: maximal velocity over aortic valve; AV dpmax: maximal pressure gradient over; AV dpmean: mean pressure gradient over aortic valve; TAPSE: tricuspid annular plane systolic excursion; AVI: aortic valve insufficiency; MVI: mitral valve insufficiency; TVI: tricuspid valve insufficiency; eGFR: estimated glomerular filtration rate; BNP: brain natriuretic peptide; HK: hematocrit; HB: hemoglobin; CK: creatine kinase.

**Table 6 ijms-25-08220-t006:** Correlation analysis (sPAP and IGF-BP2) in female gender. * *p* ≤ 0.05; ** *p* ≤ 0.01; *** *p* ≤ 0.001.

	sPAP		IGF-BP2 Baseline		IGF-BP2 Post 24 h		IGF-BP2 Post 96 h		IGF-BP2 Post 3 Months	
	r	*p*	r	*p*	r	*p*	r	*p*	r	*p*
Age	0.196	0.192	0.170	0.259	0.171	0.280	0.317	0.073	0.241	0.145
Height	0.364 *	0.019	0.249	0.095	−0.125	0.428	0.338	0.054	0.317	0.053
Weight	0.074	0.642	−0.036	0.824	−0.050	0.770	−0.124	0.514	0.006	0.973
BMI	−0.153	0.340	−0.143	0.404	−0.162	0.337	−0.261	0.164	−0.174	0.325
NYHA	−0.015	0.937	0.010	0.958	0.172	0.380	−0.083	0.693	0.398	0.060
STS score	0.264	0.077	0.259	0.083	−0.045	0.775	0.517 **	0.001	0.454 **	0.004
Diabetes mellitus	0.418 **	0.004	0.033	0.827	0.055	0.728	0.179	0.320	−0.320 *	0.050
Arterial hypertension	−0.037	0.808	−0.264	0.077	−0.65	0.682	−0.198	0.269	−0.071	0.671
CVD	−0.008	0.956	0.050	0.744	0.140	0.376	−0.064	0.724	−0.277	0.092
Previous myocardial infarction	−0.109	0.472	0.289	0.051	−0.018	0.908	0.223	0.213	−0.064	0.701
Atrial fibrillation	0.413 **	0.004	−0.124	0.412	−0.085	0.593	−0.0176	0.328	0.099	0.556
Pacemaker (before TAVR)	−0.084	0.577	−0.088	0.559	−0.028	0.862	−0.240	0.178	0.142	0.394
Malignancy	0.070	0.642	−0.134	0.374	−0.093	0.557	−0.016	0.931	−0.035	0.833
Stroke (before TAVR)	−0.099	0.513	−0.058	0.701	0.040	0.801	0.000	1.000	0.133	0.426
PAOD	−0.010	0.949	0.063	0.676	0.022	0.888	0.091	0.615	0.245	0.139
COPD	−0.239	0.109	0.149	0.322	−0.057	0.719	0.000	1.000	−0.011	0.949
LVEF	0.010	0.947	−0.265	0.082	−0.234	0.147	0.145	0.438	0.036	0.835
LVEDD	−0.186	0.258	−0.085	0.608	−0.074	0.673	−0.222	0.265	−0.180	0.333
AV Vmax	−0.037	0.823	−0.071	0.672	−0.089	0.618	0.224	0.282	0.197	0.297
AV dpmax	−0.073	0.648	−0.056	0.726	−0.149	0.373	0.203	0.282	0.251	0.146
AV dpmean	−0.003	0.984	−0.015	0.923	0.115	0.490	0.237	0.207	0.231	0.182
TAPSE	−0.461 **	0.004	−0.090	0.592	−0.343 *	0.047	−0.127	0.536	−0.139	0.457
sPAP	1.000	-	0.170	0.259	0.171	0.280	0.317	0.073	0.241	0.145
AVI ≥ II°	0.217	0.266	0.113	0.566	0.368	0.064	0.161	0.524	0.255	0.252
MVI ≥ II°	0.225	0.146	0.135	0.388	−0.014	0.934	0.278	0.137	0.198	0.254
TVI ≥ II°	0.471 **	0.001	0.079	0.616	0.131	0.426	0.221	0.241	0.140	0.422
Creatinine	0.342 *	0.020	0.151	0.315	0.113	0.475	0.345 *	0.049	0.348 *	0.032
BNP	0.199	0.245	0.171	0.319	0.270	0.135	0.347	0.077	0.329	0.066
HK	−0.264	0.076	−0.213	0.156	−0.306 *	0.049	−0.352 *	0.045	−0.259	0.117
HB	−0.213	0156	−0.163	0.279	−0.142	0.370	−0.336	0.056	−0.274	0.096
CK	0.025	0.873	−0.114	0.467	−0.043	0.795	−0.262	0.155	−0.187	0.275
IGF-BP2 baseline	0.170	0.259	1.000	-	0.422 **	0.005	0.791 ***	<0.001	0.416 **	0.009
IGF-BP2 24 h post-TAVR	0.171	0.280	0.422 **	0.005	1.000	-	0.430 *	0.012	0.335 *	0.049
IGF-BP2 96 h post-TAVR	0.317	0.073	0.791 ***	<0.001	0.430 *	0.012	1.000	-	0.586 **	0.001
IGF-BP2 3 months post-TAVR	0.241	0.145	0.416 **	0.009	0.335 *	0.049	0.586 **	0.001	1.000	-

sPAP: systolic pulmonary artery pressure; IGF-BP2: Insulin-like Growth Factor-Binding Protein 2; BMI: body mass index; CVD: cardiovascular disease; PAOD: peripheral arterial occlusive disease; COPD: chronic obstructive pulmonary disease; LVEF: left ventricular ejection fraction; sPAP: systolic pulmonary artery pressure; TAVR: transcatheter aortic valve replacement; LVEDD: left ventricular end-diastolic diameter at diastole; IVSd: interventricular septal thickness at diastole; AV Vmax: maximal velocity over aortic valve; AV dpmax: maximal pressure gradient over; AV dpmean: mean pressure gradient over aortic valve; TAPSE: tricuspid annular plane systolic excursion; AVI: aortic valve insufficiency; MVI: mitral valve insufficiency; TVI: tricuspid valve insufficiency; eGFR: estimated glomerular filtration rate; BNP: brain natriuretic peptide; HK: hematocrit; HB: hemoglobin; CK: creatine kinase.

**Table 7 ijms-25-08220-t007:** Univariate and multivariable cox regression analysis detecting 1-,3-, and 5-year mortality with dependence on male gender.

Cox Regression Analysis Male	Univariate		Multivariable	
	Hazard Ratio (95% CI)	*p*-Value	Hazard Ratio (95% CI)	*p*-Value
**1-year mortality**				
sPAP	2.228 (1.345–3.691)	0.002	1.717 (0.979–3.012)	0.059
Creatinine	738.274 (4.079–13,363.957)	0.013	4.362 (0.004–4686.232)	0.679
IGF-BP2 (baseline)	1.665 (1.185–2.340)	0.003	1.400 (1.037–1.891)	0.028
IGF-BP2 (24 h post-TAVR)	1.585 (1.136–2.211)	0.007	0.720 (0.350–1.481)	0.372
**3-year mortality**				
CVD	0.416 (0.160–1.080)	0.071	0.633 (0.227–1.764)	0.381
sPAP	1.835 (1.206–2.792)	0.005	1.450 (0.912–2.305)	0.117
Creatinine	141.341 (0.558–35,785.880)	0.080	1.142 (0.001–922.085)	0.969
IGF-BP2 (baseline)	1.593 (1.125–2.256)	0.009	1.531 (1.180–1.986)	0.001
IGF-BP2 (24 h post-TAVR)	1.529 (1.104–2.117)	0.011	0.812 (0.411–1.603)	0.548
**5-year mortality**				
STS score	1.473 (0.991–2.190)	0.055	0.758 (0.327–1.758)	0.519
sPAP	1.480 (1.041–2.105)	0.029	1.261 (0.795–2.000)	0.325
Creatinine	491.644 (4.622–52,298.685)	0.009	1.286 (0.000–570.455)	0.970
IGF-BP2 (baseline)	1.659 (1.201–2.291)	0.002	1.462 (1.075–1.988)	0.016
IGF-BP2 (24 h post-TAVR)	1.490 (1.092–2.035)	0.012	0.858 (0.354–2.077)	0.734
IGF-BP2 (96 h post-TAVR)	1.340 (0.998–1.800)	0.052	1.125 (0.504–2.512)	0.774

sPAP: systolic pulmonary artery pressure; IGF-BP2: Insulin-like Growth Factor-Binding Protein 2; CVD: cardiovascular disease.

**Table 8 ijms-25-08220-t008:** Univariate and multivariable cox regression analysis detecting 1-,3-, and 5-year mortality with dependence on male gender and PH.

Cox Regression Analysis Male + sPAP ≥ 40 mmHg	Univariate		Multivariable	
	Hazard Ratio (95% CI)	*p*-Value	Hazard Ratio (95% CI)	*p*-Value
**1-year mortality**				
Age	0.686 (0.444–1.061)	0.090	0.622 (0.385–1.005)	0.052
Creatinine	112.859 (0.458–27,780.164)	0.092	1.005 (0.001–726.721)	0.999
IGF-BP2 (baseline)	1.356 (1.027–1.791)	0.032	1.458 (1.097–1.938)	0.009
IGF-BP2 (24 h post-TAVR)	1.402 (0.980–2.007)	0.065	1.186 (0.501–2.805)	0.698
**3-year mortality**				
IGF-BP2 (baseline)	1.317 (0.991–1.750)	0.058	1.387 (1.053–1.827)	0.020
IGF-BP2 (24 h post-TAVR)	1.348 (0.945–1.922)	0.099	0.835 (0.381–1.833)	0.653
**5-year mortality**				
BMI	1.513 (0.943–2.429)	0.086	1.702 (0.966–2.999)	0.066
TVI ≥ II°	0.189 (0.043–0.823)	0.027	0.201 (0.043–0.926)	0.040
IGF-BP2 (baseline)	1.346 (1.024–1.768)	0.033	1.649 (1.162–2.339)	0.005

IGF-BP2: Insulin-like Growth Factor-Binding Protein 2; BMI: body mass index; TVI: tricuspid valve insufficiency.

**Table 9 ijms-25-08220-t009:** Univariate cox regression analysis of different IGF-BP2 baseline values detecting 1-,3-, and 5-year mortality with dependence on male gender or male gender and PH.

Univariate Cox Regression Analysis	Male		Male + sPAP ≥ 40 mmHg	
	Hazard Ratio (95% CI)	*p*-Value	Hazard Ratio (95% CI)	*p*-Value
**1-year mortality**				
IGF-BP2 (baseline) > 100 ng/mL	0.734 (0.161–3.353)	0.690	0.437 (0.094–2.033)	0.291
IGF-BP2 (baseline) > 200 ng/mL	1.416 (0.457–4.393)	0.547	1.030 (0.314–3.376)	0.962
IGF-BP2 (baseline) > 300 ng/mL	2.532 (0.761–8.422)	0.130	2.381 (0.694–8.167)	0.168
IGF-BP2 (baseline) > 400 ng/mL	3.502 (1.051–11.671)	0.041	3.071 (0.893–10.567)	0.075
IGF-BP2 (baseline) > 500 ng/mL	7.320 (2.166–24.735)	0.001	6.491 (1.828–23.044)	0.004
IGF-BP2 (baseline) > 600 ng/mL	11.258 (3.249–39.010)	<0.001	6.491 (1.828–23.044)	0.004
**3-year mortality**				
IGF-BP2 (baseline) > 100 ng/mL	0.703 (0.202–2.449)	0.580	0.383 (0.107–1.363)	0.138
IGF-BP2 (baseline) > 200 ng/mL	1.258 (0.485–3.262)	0.637	0.956 (0.346–2.639)	0.931
IGF-BP2 (baseline) > 300 ng/mL	1.633 (0.532–5.017)	0.392	1.645 (0.521–5.191)	0.396
IGF-BP2 (baseline) > 400 ng/mL	2.341 (0.760–7.207)	0.138	2.220 (0.700–7.045)	0.176
IGF-BP2 (baseline) > 500 ng/mL	5.720 (1.812–18.058)	0.003	6.491 (1.828–23.044)	0.004
IGF-BP2 (baseline) > 600 ng/mL	11.258 (3.249–39.010)	<0.001	6.491 (1.828–23.044)	0.004
**5-year mortality**				
IGF-BP2 (baseline) > 100 ng/mL	1.138 (0.340–3.805)	0.834	0.475 (0.138–1.638)	0.238
IGF-BP2 (baseline) > 200 ng/mL	1.623 (0.740–3.561)	0.227	1.182 (0.475–2.943)	0.719
IGF-BP2 (baseline) > 300 ng/mL	2.224 (0.927–5.340)	0.074	2.255 (0.850–5.985)	0.102
IGF-BP2 (baseline) > 400 ng/mL	2.678 (1.064–6.739)	0.036	2.333 (0.832–6.545)	0.108
IGF-BP2 (baseline) > 500 ng/mL	4.147 (1.392–12.360)	0.011	6.491 (1.828–23.044)	0.004
IGF-BP2 (baseline) > 600 ng/mL	11.258 (3.249–39.010)	<0.001	6.491 (1.828–23.044)	0.004

IGF-BP2: Insulin-like Growth Factor-Binding Protein 2.

## Data Availability

The datasets used and analyzed during the current study are available from the corresponding author on reasonable request.

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
