# Peer review of "Insulin-like Growth Factor-Binding Protein 2 in Severe Aortic Valve Stenosis and Pulmonary Hypertension: A Gender-Based Perspective"

_ijms, 2024, doi:10.3390/ijms25158220_

Round 1

Reviewer 1 Report

Comments and Suggestions for Authors

In the paper entitled “ Insulin-like Growth Factor Binding Protein 2 in Severe Aortic 2 Valve Stenosis and Pulmonary Hypertension: A Gender-Based 3 Perspective” the authors sought to determine the role of Insulin-like Growth Factor-Binding Protein 2 (IGF-BP2) in the clinical management of aortic stenosis-induced pulmonary hypertension. Thus, in 102 patients (56 men and 46 women) with/without  pulmonary hypertension undergoing interventional replacement of aortic valve (transcatheter aortic valve replacement – TAVR), serum levels of IGF-BP2 were determined before and after (at 24h, 96 h and at 3 months) valve replacement. Their results and correlation analysis showed a significant positive correlation between systolic pulmonary arterial pressure and IGF-BP2 across all time points suggesting a potential role of IGF-BP2 in the physiopathology of pulmonary arterial hypertension in severe aortic valve stenosis. The authors concluded that IGF-BP2 is an isolated risk factor for premature death in male alone and in male with pulmonary hypertension.  

Overall, the article is well written. The hypothesis tested is attractive; the methodology of the study is adequate. In my opinion, the article fulfill all the scientific rigors to be published. 

Reviewer 2 Report

Comments and Suggestions for Authors

I’ve read with interest the manuscript titled: “Insulin-like Growth Factor Binding Protein in Severe Aortic Valve Stenosis and Pulmonary Hypertension: A Gender-Based Perspective” by Boxhammer and associates.

I have the following comments to make:

  1. Your definition of pulmonary hypertension, based on echographic assessment of systolic pulmonary pressure values is prone to failure at multiple levels. You should have used a single echographic criterion, such as TRV, in order to limit the amount of errors introduced at every step. Your usage  of RAP is dependent on the filling of the patient.
  2. Your RAP assumptions had the value of 3,8 or 15 mm Hg. These discrete values introduced into the formula you used (4 × TRVmax2) + RAP can lead to significant alterations in your results and bias your data as well as shift the category in which the patients was used in your study.
  3. You mention in Lines 113-114 that “An sPAP ≥ 40 mmHg was considered the cut-off value for diagnosing PH based on current literature”. Please provide citations. PAH is classified by ESC using mean PA values
  4. Please provide in Material and Method section information about your method of defining COPD. This is crucial when talking about pulmonary hypertension.
  5. There is currently no information about spirometry values in your cohort. Please provide a separate table with those values, as well as gender specific differences in those values. 
  6. How did you choose your Cox regression analysis covariates? Specifically, why didn’t you include sPAP values in this analysis, either as a continuous variable or as a dichotomous one?
  7. There are no information about the size of the TAVR valves used. 
  8. No information about gradients at discharge and 3 months after TAVR implantation. Women tend to have smaller annular aortic annuli and thus smaller implanted Evolut valves than men. It is conceivable that the gender difference you notice at 3 months (higher IGF-BP2 in women) may be due to higher gradients in this population.
  9. No information about paravalvular leaks. In general, you provide absolutely no echographic follow-up about the fate of your TAVR valves and explain the differences at 3 months by gender. This conclusion can only be taken in the absence of other explanations, which you fail to address in your study.

Reviewer 3 Report

Comments and Suggestions for Authors

The manuscript is well written, the methodology is clear and the results presented in a balanced way. Some minori points are reported below: 

#1. Tables 1, 4-5 are too long and should be separate into different ones to facilitate the Readers in exploring all the results of the analysis. 

#2. In figures, please avoid the use of symbols and adopt the terms "overall population, male or female groups".

#3.  How many patients have been excluded from the analysis and which were the main exclusion criteria. 

#4. How long was the recruitment period?

#5. Apparently one singole physician (or sonographer) performed all the echocardiographic assessments before and after TAVR. This aspect, as well as the single center, descriptive nature of the study must be acknowledged both in the abstract and in the conclusions. 

Round 2

Reviewer 2 Report

Comments and Suggestions for Authors

The authors have addressed all the issues raised in the initial review, with the exception of the lack of functional pulmonary testing. The lack of these data means that there is no information on the pulmonary functional status, which can decisively influence the presence of pulmonary hypertension. This limitation should also be added in the relevant section in your manuscript.

Author Response

Comment: The authors have addressed all the issues raised in the initial review, with the exception of the lack of functional pulmonary testing. The lack of these data means that there is no information on the pulmonary functional status, which can decisively influence the presence of pulmonary hypertension. This limitation should also be added in the relevant section in your manuscript.

Response: Thank you very much for pointing this out. We included the lack of functional pulmonary tests in the limitation section of our manuscript.